# Using Fractal Dimension Analysis with the Desikan–Killiany Atlas to Assess the Effects of Normal Aging on Subregional Cortex Alterations in Adulthood

**DOI:** 10.3390/brainsci11010107

**Published:** 2021-01-14

**Authors:** Chi-Wen Jao, Chi Ieong Lau, Li-Ming Lien, Yuh-Feng Tsai, Kuang-En Chu, Chen-Yu Hsiao, Jiann-Horng Yeh, Yu-Te Wu

**Affiliations:** 1Institute of Biophotonics, National Yang-Ming University, Taipei 112, Taiwan; c3665810@ms24.hinet.net (C.-W.J.); chiieong.lau@ndcn.oxon.org (C.I.L.); 2Department of Research, Shin Kong Wu Ho-Su Memorial Hospital, Taipei 111, Taiwan; 3School of Medicine, College of Medicine, Fu Jen Catholic University, New Taipei 242, Taiwan; yuhfeng.tsai@gmail.com; 4Dementia Center, Department of Neurology, Shin Kong Wu Ho-Su Memorial Hospital, Taipei 111, Taiwan; 5Applied Cognitive Neuroscience Group, Institute of Cognitive Neuroscience, University College London, London WC1N 3AZ, UK; 6Department of Neurology, University Hospital, Taipa 999078, Macau; 7Department of Neurology, Shin Kong Wu Ho-Su Memorial Hospital, Taipei 111, Taiwan; M002177@ms.skh.org.tw; 8Department of Neurology, School of Medicine, College of Medicine, Taipei Medical University, Taipei 110, Taiwan; 9Department of Diagnostic Radiology, Shin Kong Wu Ho Su Memorial Hospital, Taipei 111, Taiwan; leo_eraser@yahoo.com.tw; 10Department of Gastroenterology, Shin Kong Wo Ho-Su Memorial Hospital, Taipei 111, Taiwan; alkec55@gmail.com; 11Health Management Center, Shin Kong Wo Ho-Su Memorial Hospital, Taipei 111, Taiwan; 12Brain Research Center, National Yang-Ming University, Taipei 112, Taiwan

**Keywords:** normal aging, fractal dimension, Desikan–Killiany Atlas, prediagnosis

## Abstract

Normal aging is associated with functional and structural alterations in the human brain. The effects of normal aging and gender on morphological changes in specific regions of the brain are unknown. The fractal dimension (FD) can be a quantitative measure of cerebral folding. In this study, we used 3D-FD analysis with the Desikan–Killiany (DK) atlas to assess subregional morphological changes in adulthood. A total of 258 participants (112 women and 146 men) aged 30–85 years participated in this study. Participants in the middle-age group exhibited a decreased FD in the lateral frontal lobes, which then spread to the temporal and parietal lobes. Men exhibited an earlier and more significant decrease in FD values, mainly in the right frontal and left parietal lobes. Men exhibited more of a decrease in FD values in the subregions on the left than those in the right, whereas women exhibited more of a decrease in the lateral subregions. Older men were at a higher risk of developing mild cognitive impairment (MCI) and exhibited age-related memory decline earlier than women. Our FD analysis using the DK atlas-based prediagnosis may provide a suitable tool for assessing normal aging and neurodegeneration between groups or in individual patients.

## 1. Introduction

Normal aging is associated with functional and structural alterations in the human brain. An accurate description of age-related changes in the brain is required to understand the neurobiological foundation of cognitive changes in healthy aging [1]. Studies have reported the use of structural magnetic resonance imaging (MRI) in detecting functionally relevant brain changes and assessing cortical morphological changes during normal aging [2,3,4,5,6]. Globally, human brain volume was found to decrease with age in the gray matter (GM) and white matter (WM) compartments [7,8]. The mesolimbic, temporal, and frontal regions show greater vulnerability to the effects of aging [9,10,11,12]. Accelerating or decelerating changes may occur in different cortical subregions in response to aging. Accelerating and decelerating changes with increasing age were reported in the temporo-occipital, prefrontal, and anterior cingulate cortices, respectively [13]. A study on normal aging effects on cortical thickness reported a rapid decline in the parietal and insula regions, moderate decline in the frontal and occipital lobes, and slow decline in the cingulate and temporal lobes [14]. The rate of changes in the cortical region is considered a predictor of aging effects [14,15,16,17]. Atrophy of the medial temporal lobe (MTL) of the brain is key to memory function in old age [15], and the atrophy rate of MTL can be a predictor of aging [16]. Regional decline in the medial occipitotemporal, inferior, and middle temporal lobes was reported to predict Alzheimer’s disease [17]. Age-dependent volume loss in the cortical thickness of the temporal lobe and hippocampus is a biomarker for memory loss and cognitive decline in normal aging, and can be used to differentiate dementia from healthy aging [14].

Studies have suggested that men and women showed different brain atrophy with aging [18,19]. Moreover, the degree of brain atrophy was milder in women than in men with aging; however, the opposite results have also been reported [18]. After controlling for total percent brain volume change, men showed greater relative regional brain reduction than women in bilateral precentral gyri, bilateral paracingulate gyri, and bilateral supplementary motor cortices [20]. Sex differences may affect lateralization of brain atrophy with aging. Elderly men exhibited more atrophy in the left hemisphere, whereas the atrophy was symmetric in elderly women [19]. Besides normal aging, sex hormones also have a significant influence on alterations of brain structure, behavior, and cognitive functioning [20,21,22,23]. Postmenopausal women showed significantly decreased gray matter volumes of the insula, putamen, parahippocampal gyrus, amygdala, and anterior cingulate gyrus. These decreased regions were associated with decreased functional activity during visual sexual arousal in postmenopausal women [24]. For females across the menstrual cycle, intrinsic fluctuations in progesterone were associated with volumetric changes in CA2/3 of the middle temporal lobe, as well as the entorhinal, perirhinal, and parahippocampal cortex [25].

Quantifying alterations in complex structures, such as cortical sulci and gyri, is challenging. Morphological features, including GM volume [26], cortical thickness [27], and surface area [28], are used to quantify structural alterations in the aging brain through MRI. Compared with cortical thickness, these conventional volumetric methods may reveal larger variances and more of a sex effect [29]. Due to the large volumetric variance between individuals, including healthy controls, it is difficult to define atrophy or volumetric change for a single participant based on volumetric measurements [29]. Fractal dimension (FD) analysis, proposed by Mandelbrot [30], is superior to conventional volumetric methods in quantifying structural changes in cortical WM and GM because of the smaller variance and sex effect [29]. The FD can be used to quantify the cortical complexity of objects as a single numerical value, and may serve as a quantitative measure for accurately describing the cortical complexity of cerebral folding [29,31,32,33].

Brain atrophy can occur due to natural aging. Morphological alterations in functionally specific cortical regions may reveal the associated functional impairment or decay. Studies on morphological changes in cortical subregions during normal aging have been limited. In this study, we assessed the effect of normal aging on morphological alteration of the subregion of the cortex across different stages of adulthood. A total of 258 participants (112 women and 146 men) aged 30–85 years old participated in this data-driven study. All participants were grouped into three age groups of adulthood, namely young (<45 years old), middle (45–60 years old), and old (>60 years old). For each participant, the cerebral cortex was parcellated into 68 subregions according to the Desikan–Killiany (DK) atlas [34]. Three-dimensional box-counted (3D-FD) analysis was used to assess morphological changes in cerebral lobes and parcellated subregions, and the measured 3D-FD values were compared between different age stage groups and between sexes. We hypothesized different degrees of morphological alterations in cortical subregions across different stages of adulthood to reveal the sex effect on morphological changes in normal aging. First, we assessed whether FD values were a precise and superior predictor for aging and sex effect. Second, we hypothesized that those morphological alterations in the subregions manifest as functional impairment across different stages of adulthood and the sex effect on normal aging.

## 2. Materials and Methods

### 2.1. Participants

A total of 258 healthy adults (30–85 years old, 112 women and 146 men) participated in this study. All of the participants were recruited from the Department of Neurology at the Shin Kong Wu Ho-Su Memorial Hospital from 2015 to 2019. All participants were confirmed by neurologists to have no disease of the central nervous system and no neurological abnormalities during the study. Referring to the age stages of previous studies [35,36,37], in this study, the young, middle, and old adulthood groups included individuals less than 45, from 45 to 60, and more than 60 years old, respectively. The study protocol (code: 202000104R) was approved by the Institutional Review Board of the Shin Kong Wu Ho-Su Memorial Hospital on 9 April 2020. Table 1 summarizes the demographic data of each study group.

### 2.2. Image Acquisition and DK Cortical Atlas Parcellation

Figure 1 illustrates the procedure for image processing. First, axial MRI encompassing the cerebrum and cerebellum was performed using the 3-T Vision scanner (Philips Medical Systems, The Netherlands). The participants were scanned using a circularly polarized head coil to obtain T1-weighted images (repetition time, 8.875 ms; echo time, 4.079 ms; matrix size, 384 × 384; 0.9-mm sagittal slices; field of view, 256 × 256 mm; voxel size, 0.67 × 0.67 × 0.9 mm^3^). Second, each structural MRI dataset was normalized to a pre-segmented and validated volumetric template by using DiffeoMap, and each normalized image volume was then segmented into GM, WM, and cerebrospinal fluid in the native space by using the SPM8 toolbox (Welcome Center for Human Neuroimaging, UCL Queen Square Institute of Neurology). Third, the cortex of the GM was parcellated and aligned into 68 distinct regions of interest (ROIs) with structures in the DK cortical atlas by using the FreeSurfer toolbox (MATLAB R2019b; MathWorks, Natick, MA, USA). Table 2 presents a summary of the cortical parcellation regions and their index numbers, according to the DK atlas, analyzed using FreeSurfer v5.0.2. In the DK atlas, 34 cerebral subregions (ROIs) are present in each hemisphere. We rearranged the ROIs on the basis of lobes and placed them in the following order: frontal lobe (1–20, odd and even numbers for the left and right hemispheres, respectively), limbic lobe (21–32), temporal lobe (33–46), parietal lobe (47–58), and occipital lobe (59–68). Fourth, 3D-FD was used to measure the FD value of each parcellated region and lobes for each study group. Subsequently, FD values between different age groups within the same sex and of the same age group between sexes were compared.

### 2.3. Three-Dimensional FD Measurement of Parcellated Regions

A fractal is an irregular object in which part of the compartment resembles the overall shape. Kiselev et al. demonstrated that the cerebral cortex had the self-similar fractal structure and its FD value was 2.80 ± 0.05 [38]. Shan et al. suggested that the FD index is a compact measure of structural complexity, and can be used as a summary index of structure irregularity [39]. The FD has been extensively used to investigate morphological changes caused by neurological diseases in the cerebral cortex [16,17,18]. For assessing cortical alteration in aging, the FD has been reported that was more strongly related to age than other cortical measures [40]. Recently, Marzi et al. demonstrated that the FD revealed strong associations with age and yielded the most accurate machine learning models for individual age prediction [41].

The FD is a quantitative indicator of object complexity (or irregularity), which can be calculated on the basis of different metrics, namely the Hausdorff dimension and box-counting dimension. In this study, we used the box-counting method for additional morphological estimation to evaluate the complexity (or irregularity) of the human brain. We extended the 2D box-counting method in the HarFA software to a 3D method for measuring the FD values of the human cortical brain [30]. According to the power law relationship, the FD of a fractal is defined as N(r) ∝ r ***^−FD^*,** where $N\left(r\right)$ is the total number of boxes required to completely cover the fractal object with boxes of a different size, r, and can be rewritten as: $\log\left(N\left(r\right)\right)=\mathrm{FD}\cdot \log\left(\frac{1}{r}\right)+k$; thus, the value of FD can be estimated from the slope of the line between $\log\left(N\left(r\right)\right)$ and $\log\left(1/r\right)$).

A box of a given size was filled with one whenever a part of the object appeared within a box, and the total number of boxes required to completely cover the object was counted. The total number of boxes changed when a difference box size, r, was used [29]. The process was repeated, and a log–log scatter plot of ln N(r) versus ln(1/r) was plotted (see Figure 2). The value of the FD was obtained from the slope of the fitted line using the MATLAB 2020a software (MathWorks, Inc., Natick, MA, USA). Because FD analysis is based on a logarithmic scale, a more complex cerebral structure has a higher FD value, whereas a degeneration of the cerebral structure is indicated by a decrease in the FD value [6]. Hence, the FD can help evaluate minor morphological changes in the brain cortex in normal aging.

Notably, the range of box sizes must be carefully selected, because the cerebral cortex is self-similar only in a certain scale range and not strictly purely fractal. Studies have reported that a smaller box size produces a higher FD value and a larger box size produces a smaller FD value, and determining the range of box sizes corresponding to the linear part of the curve to retain the fractal characteristics of the cerebral GM volumetric images is important for FD estimation [29,42]. In other words, analysis of a single slope with different box sizes can determine the range that achieves the highest slope correlation coefficient (R^2^) [29,42]. Figure 2 illustrates the procedure for selecting the range of the box size. The highest correlation coefficient (R^2^ = 0.995) was obtained at an optimal box size ranging between two and eight pixels. Linear regression analysis was performed on the linear part of the curve to obtain the FD value (red line in Figure 2), and the FD value obtained from the slope of the fitted line in this range (box size: two to eight pixels) was 2.2962.

### 2.4. Statistical Analysis

We used a two-tailed *t* test to determine whether a significant difference existed between different age groups of the same sex and between two sexes of the same age groups regarding the 3D-FD values of each cortical region. The significance of the results was based on the false discovery rate (FDR)-corrected *p* = 0.05 [43]. These analyses were conducted using the Statistics Toolbox in MATLAB 2020a version.

## 3. Results

### 3.1. Women and Men Exhibited More Cortical Lateralization in Young and Middle Adulthood, Respectively

Table 3 and Table 4 summarize FD values of subregions at different ages for women and men, respectively. During young adulthood, women exhibited more cortical lateralization than did men. Young women exhibited significantly smaller FD values for nine and five subregions in the left and right hemispheres, respectively. Young men exhibited less cortical lateralization and had significantly smaller FD values for three and two subregions in the left and right hemispheres, respectively. With progression to middle age, men exhibited increased cortical lateralization; 24 subregions had significantly asymmetrical differences, and smaller FD values were observed mainly in the frontal, temporal, and parietal lobes of the left hemisphere (left/right = 15/9).

### 3.2. Aging Affects the Bilateral Frontal and Left Temporal Lobes Early

Table 5 summarizes significant morphological alterations (decrease in FD values) in the subregions between different age groups: young-age group vs. middle-age group and middle-age group vs. old-age group. Figure 3a illustrates the location of these subregions from young to middle age and Figure 3b from middle to old age. In Figure 3a, in the young to middle-age group (including men and women), three subregions in the left hemisphere, including the frontal and temporal lobes, and two subregions in the right hemisphere, including the frontal and limbic lobes, exhibited significantly decreased FD values. For the middle-age group, cortical complexity began decreasing in the bilateral frontal lobes first. Furthermore, the left middle orbitofrontal of the middle-age group had a larger FD value than that of the young-age group.

As adulthood progressed to old age, more subregions underwent morphological alterations, which occurred in all lobes of both hemispheres. The temporal, parietal, and left limbic lobes exhibited a significant decrease in cortical complexity from middle to old age. Notably, the old-age group exhibited a decrease in FD values in the bilateral rostral middle frontal, middle temporal, superior temporal, transverse temporal, superior parietal, and lateral occipital regions. Furthermore, from young to middle age, the rostral middle frontal region of the left hemisphere exhibited a decrease in FD values, which continued to decrease as old age progressed. Moreover, the old-age group exhibited increased FD values in the occipital lobe, including the bilateral pericalcarine and right cuneus subregions.

### 3.3. Men Exhibited a Decrease in Cortical Complexity Earlier and in More Subregions than Did Women across Two Age Periods of Adulthood

We assessed the effect of sex differences on normal aging between men and women. Table 6 summarizes the significantly altered subregions between two different age periods for men and women. For women, Figure 4a,b demonstrates the location of the altered regions from young to middle age and from middle to old age, respectively. Figure 5a,b demonstrates the location of the altered regions of these two age periods for men. Overall, men exhibited a more significant decrease in cortical complexity earlier in different subregions than did women across different stages of adulthood. Middle-age women exhibited six subregions with significantly smaller FD values (L/R = 3/3) than did young-age women. Middle-age men exhibited 14 subregions with significantly decreased FD values (8/6 = L/R) from young to middle age. As indicated in Table 6, middle-age men exhibited more subregions with significantly smaller FD values than did middle-age women in the left parietal and right frontal lobes.

The effect of sex differences on normal aging continued as old age progressed, and men exhibited more morphological alterations in the subregions than did women. From middle to old age, women exhibited more significantly decreased FD values for six cortical subregions (L/R = 3/3), mainly including the frontal (left rostral middle frontal and bilateral caudal middle frontal) and occipital (bilateral lateral occipital) lobes. Middle- to old-age men exhibited more significantly decreased FD values for nine subregions. As illustrated in Table 6, men in the old-age group exhibited more decreased FD values mainly in the right temporal lobe, including the middle, superior, and transverse temporal lobes.

## 4. Discussion

In this study, we proposed a box size selection procedure for measuring FD, and assessed the effects of sex difference and aging on cerebral cortical complexity across different stages of adulthood. We parcellated the cerebral cortex into 68 subregions (ROIs) according to the DK atlas. FD values were measured for each subregion and compared between different age periods and sexes.

We found the following: (1) the optimum box size for measuring the FD of the cerebral cortex ranged between two and eight pixels, (2) women exhibited more asymmetrical symptoms of cortical complexity in young age, whereas men exhibited more such asymmetrical symptoms in middle age, (3) in the middle-age group, morphological alteration began in the lateral frontal lobes and then spread mainly to the temporal and parietal lobes from middle to old age in both men and women, (4) from young age to middle age, men exhibited an earlier and more significant decrease in FD values in the right frontal and left parietal lobes than did women, and (5) men exhibited a larger decrease in FD values in subregions on the left than those on the right, whereas women exhibited a decrease in the lateral subregions, when comparing hemispheres.

### 4.1. Measuring FD Using the Cerebral Atlas Is a Superior Measure for Assessing Subregional Morphological Alterations in Normal Aging and Neurodegenerative Disease

FD measurement is a sensitive method for describing the atrophy of cortical structures in neurologic and psychiatric conditions [44]. Studies using voxel-based morphometry (VBM) have found a globally reduced GM volume, but inconsistent results have emerged in identifying regional changes in neurodegenerative disease [29,45]. The VBM method is a whole-brain technique that employs spatial normalization and tissue classification to discern tissue concentration differences in structural MRI [29]. VBM is used for qualitative analysis, and is suitable for group-based comparison, but not for individual assessment [29]. In this study, the 3D-FD measure obtained using the DK atlas can be used for the complete anatomical analysis of the brain and can detect a single-person ROI for cortical alteration in specific regions. We speculate that our method is better at predicting the risk of developing a degenerative disease before the symptoms appear, which is significant for developing effective and personalized medicine.

### 4.2. Men Exhibited More Aging Effects on Cerebral Morphological Changes than Women

GM thickness is mostly used for assessing normal aging or cortical neurodegeneration. Throughout adulthood, cortical thickness is most pronounced laterally in the temporal, superior, and caudal middle frontal cortices and medially in the posterior cingulate cortex [13]. Mild cognitive impairment (MCI) is a common condition in elderly people [46]. Cortical thinning in the caudal middle frontal region was reported to be associated with executive dysfunction in MCI [47]. Among the elderly populations, older men may be at a higher risk of developing MCI, which is often a precursor to Alzheimer’s disease, earlier in life than do older women [48]. The higher prevalence of MCI in men suggests that women transition from normal cognition directly to dementia at a later age, but more abruptly [48]. Our findings reveal that sex differences may affect the occurring time and disease progression in MCI. As illustrated in Table 6, men exhibited a significant decrease in FD values earlier in the right caudal middle frontal cortices from young to middle age. Women exhibited a significant decrease in FD values later in the bilateral caudal middle frontal cortices from middle to old age, which may explain the effect of sex difference on MCI.

The temporal lobe is a critically functional part of the cerebral cortex [49]. Atrophy of the MTL may cause memory function decline and memory complaints in old age [15]. Structural imaging studies have found that smaller MTL volumes are associated with lower memory performance [39]. During normal aging, MTL volumes and right frontal-lobe function have a positive association with variability in memory performance among the elderly, suggesting a two-stage model of memory decline in aging [50]. As seen in Table 6, men exhibited a significant decrease in FD values in many subregions of the right frontal lobe from young to middle age, which then decreased significantly in the right middle temporal cortices from middle to old age. Our finding may explain the two-stage model of memory decline in aging. Furthermore, our findings suggest that the tendency of age-related memory decline is greater in men than in women.

### 4.3. Measuring FD Using the DK Atlas to Develop a Prediagnosis System for Assessing Aging and Cortical Neurodegenerative Disease

Brain age estimation from anatomical features has been attracting considerable interest in neuroscience [51,52]. Studies have revealed that normal aging follows a specific pattern, which can enable the prediction of the age of the human brain from its degeneration status [53]. A cortical features-based predictor may help identify individuals at an increased risk of brain disease [53]. We proposed a novel method for measuring FD using a DK atlas-based pre-diagnosis system for assessing normal aging and cortical neurodegenerative disease. This pre-diagnosis system can be used for intergroup comparison or for assessing individual participants.

Figure 6a,b illustrates the scatter plots of the pre-diagnosis system for middle-age men (Figure 6a) and women (Figure 6b). In Figure 6a, a blue star represents each cortical subregion for men in the middle-age group; the distance from the center to the blue star is the mean FD value of each cortical subregion. The left and right parts of the figure illustrate the subregions of the left and right hemispheres, respectively. Each subregion is arranged in the order mentioned in Table 2. The subregions of the frontal lobe (1–19 and 2–20) are depicted by red lines, subregions of the limbic cortex (21–31 and 22–32) are depicted by green lines, and so forth. The distance from the center to the otter contour blue line is equal to the blue star FD value plus its standard deviation, and the distance from the center to the inner contour blue line is equal to the blue star FD value subtracted by its standard deviation. The red dots indicate the FD values of each subregion of one participant (participant 37 from men in the middle-age group, Figure 6a). As in Figure 6a, the subregions with significantly smaller FD of the test participant were labeled (PaH, IT, ST, TPol, MOrF, … in Figure 6a). We defined if the FD value of each subregion of the test participant was smaller than his/her peer group, with the FD value of the blue star (mean FD value of the index of the identical subregion) subtracted by the standard deviation; these significantly smaller subregions are labeled as the susceptive of abnormal subregions. In Figure 6a, the labeled regions indicate that participant 37 from the men in the middle-age group had significant morphological alterations in his left temporal and right frontal lobes compared with his age-matched peers. Figure 6b shows significantly smaller subregions in participant 37 from the women in the middle-age group compared with her age-matched peers. Participant 37 from the women in the middle-age group exhibited four subregions of significantly smaller FD values, whereas participant 37 from the men in the middle-age group exhibited nine subregions of significantly smaller FD values. Thus, we may suggest that our proposed FD values using the DK atlas-based pre-diagnosis system is a suitable tool for comparing and pre-assessing symptoms of normal aging or neurodegeneration between groups or in individual participants.

However, there are some limitations in the present study. The phases of the menstrual cycle, taking oral contraceptives, and the state of the menopause period of each female subject was not inquired and investigated. These data are causal for assessing sex difference of normal aging. Hence, in this study, the effect of hormone states on subcortical alteration along normal aging cannot be assessed, especially for the middle-age female group, who are in stages of menopausal transition.

## 5. Conclusions

We performed 3D-FD analysis using the DK atlas to assess subregional morphological changes across different stages of adulthood. Our method could be effective in predicting the risk of developing a degenerative disease before symptoms appear. Men exhibited more of a decrease in FD values in the subregions on the left than in those on the right, whereas women exhibited more of a decrease in the lateral subregions. Men exhibited an earlier and more significant decrease in subregion FD values, mainly in the right frontal and left parietal lobes. Older men were at a higher risk of developing MCI and age-related memory decline earlier than women. Our pre-diagnosis system may provide a suitable tool for comparing symptoms of normal aging and neurodegeneration between groups or in individual participants.

## Figures and Tables

**Figure 1 brainsci-11-00107-f001:**
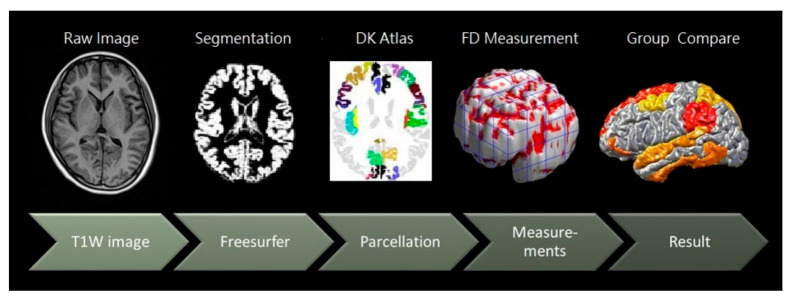
The image processing procedure.

**Figure 2 brainsci-11-00107-f002:**
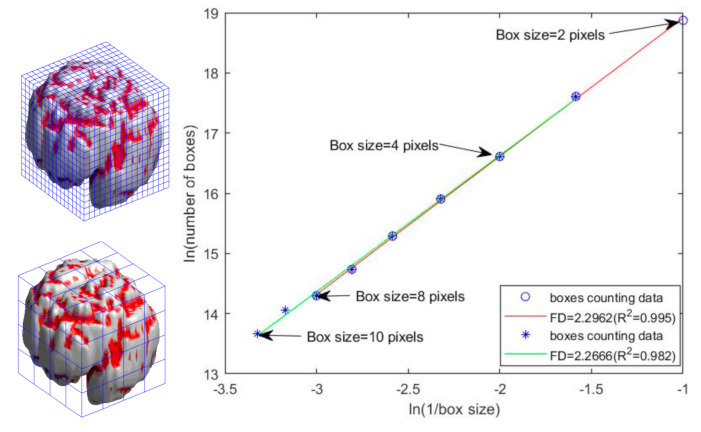
Computational procedure for FD estimation. Left: the smaller the box size, the larger the number of boxes counted, and vice versa. Right: log–log scatter plot, where the x and y axis denote the inverse of the box size and the number of boxes in the logarithmic scale, respectively.

**Figure 3 brainsci-11-00107-f003:**
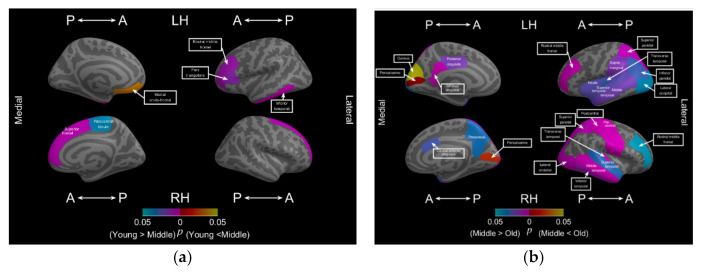
Significantly decreased FD values in cortical subregions between different age period; (**a**) significant decrease in FD values in cortical subregions from young to middle age; (**b**) significant decrease in FD values in cortical subregions from middle to old age.

**Figure 4 brainsci-11-00107-f004:**
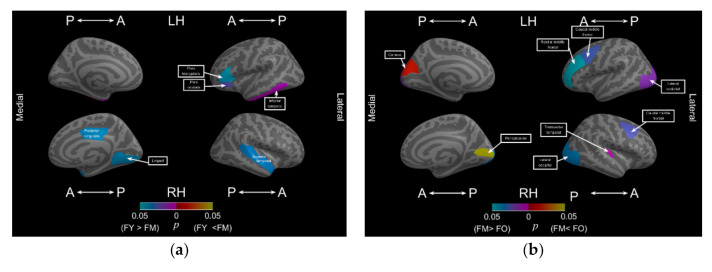
Significant decrease in FD values in the subregions in women between different age stages. (**a**) Significant decrease in FD values in the subregions in women in the young-age group compared with that in women in the middle-age group; (**b**) significant decrease in FD values in the subregions in women in the middle-age group compared with that in women in the old-age group.

**Figure 5 brainsci-11-00107-f005:**
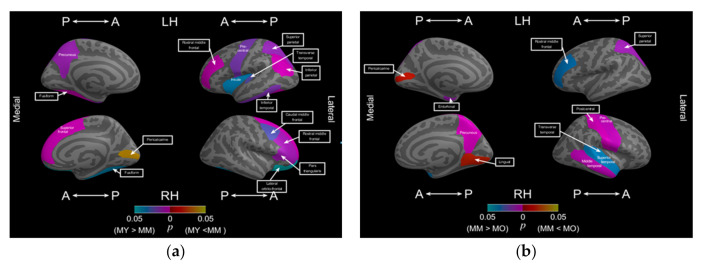
Significant decrease in FD values in the subregions in men between different age stages. (**a**) Significant decrease in FD values in the subregions in men in the young-age group compared with that in men in the middle-age group; (**b**) decrease in fractal dimension (FD) values in the subregions in men in the middle-age group compared with that in men in the old-age group.

**Figure 6 brainsci-11-00107-f006:**
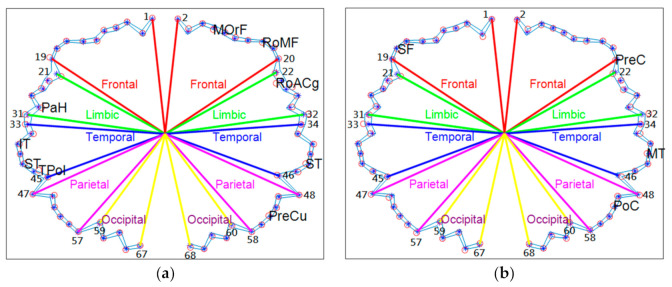
Pre-diagnosis system based on FD with the Desikan–Killiany atlas measuring (**a**) participant 37 from the men in the middle-age group vs. men in the middle-age group; (**b**) participant 37 from the women in the middle-age group vs. women in the middle-age group. In each figure, the blue star represents each cortical subregion in the middle-age group; the distance from the center to the blue star is the mean FD value of each cortical subregion. The red dots indicate the FD values of each subregion of one test participant. If the FD value of each subregion of the test participant was smaller than his/her peer group, the FD value of the blue star (mean FD value of the index of the identical subregion) subtracted by the standard deviation, then these significantly smaller subregions are labeled as the susceptive of abnormal subregions.

**Table 1 brainsci-11-00107-t001:** Demographic data across different stages of adulthood in men and women.

Group	Young (Number/Mean Age ± std)	Middle	Old
Female	30/39.8 ± 3.6	41/53.94 ± 4.44	41/67.23 ± 5.34
Male	36/38.93 ± 7.22	57/53.61 ± 4.25	53/67.8 ± 4.52
Total	66/39.45 ± 6.08	98/53.74 ± 4.32	94/67.35 ± 4.87

**Table 2 brainsci-11-00107-t002:** Regions in the Desikan–Killiany (DK) atlas.

Frontal	ROI	Abbreviation	Temporal	ROI	Abbreviation
1, 2	Caudal middle frontal	CMF	33, 34	Bankssts	B
3, 4	Frontal pole	FPol	35, 36	Entorhinal	En
5, 6	Lateral orbitofrontal	LOrF	37, 38	Inferior temporal	IT
7, 8	Medial orbitofrontal	MOrF	39, 40	Middle temporal	MT
9, 10	Pars opercularis	Op	41, 42	Superior temporal	ST
11, 12	Pars orbitalis	Or	43, 44	Temporal pole	TPol
13, 14	Pars triangularis	Tr	45, 46	Transverse temporal	TrT
15, 16	Rostral middle frontal	RoMF	Parietal		
17, 18	Superior frontal	SF	47, 48	Inferior parietal	IP
19, 20	Precentral gyrus	PreC	49, 50	Paracentral	PaC
Limbic			51, 52	Postcentral	PoC
21, 22	Caudal anterior cingulate	CACg	53, 54	Precuneus	PreCu
23, 24	Rostral anterior cingulate	RoACg	55, 56	Superior parietal	SP
25, 26	Isthmus cingulate	IstCg	57, 58	Supra marginal	SM
27, 28	Insula	Ins	Occipital		
29, 30	Parahippocampal	PaH	59, 60	Pericalcarine	PerCa
31, 32	Posterior cingulate	PoCg	61, 62	Fusiform	Fu
			63, 64	Cuneus	Cu
			65, 66	Lateral occipital	LO
			67, 68	Lingual	Lg

**Table 3 brainsci-11-00107-t003:** Subregion fractal dimension (FD) values in women of different ages.

Lobes	Index	FY (L)	FY(R)	FM(L)	FM(R)	FO(L)	FO (R)
Frontal Lobe	CMF	2.0806 *	2.1507	2.0718 ^**^	2.1259	2.0650 ^***^	2.1163
	FPol	2.2815	2.2993	2.2933	2.2886	2.2806	2.2731
	LOrF	2.2873	2.2770	2.2848	2.2636 ^**^	2.2768	2.2573 ^***^
	MOrF	2.2125 *	2.2448	2.2373	2.2379	2.2240	2.2380
	Op	2.1693 *	2.2057	2.1624 ^**^	2.1919	2.1565 ^***^	2.2012
	Or	2.2693	2.2169 *	2.2542	2.2180 ^**^	2.2451	2.2114 ^***^
	Tr	2.1662	2.1879	2.1351 ^**^	2.1707	2.1368 ^***^	*2.1803*
	RoMF	2.2512	2.2736	2.2306 ^**^	2.2688	2.2261 ^***^	2.2645
	SF	2.3137	2.3069	2.3141	2.3083	2.3114	2.3028
	PreC	2.1178	2.0897	2.1295	2.1084	2.1195	2.0936
Limbic Lobe	CACg	2.4088	2.3993	3.4011	2.3946	2.3920	2.3916
	RoACg	2.4058	2.4057	2.4055	2.4008	2.4012	2.3980
	IstCg	2.0447	2.0605	2.0154 ^**^	2.0493	2.0342	2.0553
	Ins	2.2366	2.2221	2.2335	2.2275	2.2250	2.2319
	PaH	2.1695	2.1575	2.1519	2.1594	2.1545	2.1463
	PoCg	2.0932	2.0939	2.0817	2.0745	2.0891	2.0883
Temporal Lobe	B	2.2901	2.2803	2.2914	2.2796 ^**^	2.2823	2.2724
	En	2.3350	2.3147 *	2.3171	2.3104	2.3242	2.3023 ^***^
	IT	2.3038	2.3158	2.2965 ^**^	2.3096	2.2873 ^***^	2.3018
	MT	2.0800	2.0214 *	2.0606	2.0210 ^**^	2.0503	2.0173 ^***^
	ST	2.3170 *	2.3389	2.3140 ^**^	2.3255	2.3074 ^***^	2.3242
	TPol	2.1664	2.1766	2.1869	2.1846	2.1940	2.1776
	TrT	2.0390	1.9603 *	2.0298	1.9688 ^**^	2.0157	1.9320 ^***^
Parietal Lobe	IP	2.3903 *	2.4234	2.3947 ^**^	2.4201	2.3891 ^***^	2.4181
	PaC	2.1032	2.0912	2.1002	2.0989	2.0834	2.0867
	PoC	2.2889	2.2679 *	2.2791	2.2703	2.2844	2.2639 ^***^
	PreCu	2.1777 *	2.2242	2.1771	2.1932	2.1542 ^***^	2.1844
	SP	2.3139 *	2.3409	2.3158	2.3271	2.3065 ^***^	2.3306
	SM	2.3508	2.3583	2.3498	2.3484	2.3436	2.3402
Occipital Lobe	PerCa	2.3726	2.3717	2.3784	2.3589 ^**^	2.3718	2.3541 ***
	Fu	2.0687 *	2.1364	2.0853 ^**^	2.1352	2.1085 ^***^	2.1379
	Cu	2.3672	2.3688	2.3633	2.3682	2.3494	2.3557
	LO	2.2647	2.2663	2.2447	2.2459	2.2479	2.2506
	Lg	2.0355 *	2.0975	2.0519	2.0648	2.0815	2.0935
Number of significant smaller sub-regions		**9**	5	9	6	10	7

* Significantly smaller FD value (*p* < 0.05, FDR corrected) subregion of female young age group; ** significantly smaller FD value (*p* < 0.05, FDR corrected) subregion of female middle age group; *** significantly smaller FD value (*p* < 0.05, FDR corrected) subregion of female old age group. B, bankssts; CACg, caudal anterior cingulate; CMF, caudal middle frontal; Cu, cuneus; En, entorhinal; FPol, frontal pole; Fu, fusiform; IP, inferior parietal; IT, inferior temporal; Ins, insula; IstCg, isthmus cingulate; LO, lateral occipital; LOrF, lateral orbitofrontal; Lg, lingual; MOrF, medial orbitofrontal; MT, middle temporal; Or, pars orbitalis; Op, pars opercularis; Tr, pars triangularis; PaC, paracentral; PaH, parahippocampal; PerCa, pericalcarine; PreC, precentral gyrus; PreCu, precuneus; PoC, postcentral; PoCg, posterior cingulate; RoACg, rostral anterior cingulate; RoMF, rostral middle frontal; SF, superior frontal; SM, supra marginal; SP, superior parietal; ST, superior temporal; TPol, temporal pole; TrT, transverse temporal; FY(L), female young age left hemisphere; FY(R), female young age right hemisphere; FM(L), female middle age left hemisphere; FM(R), female middle age right hemisphere; FO(L), female old age left hemisphere; FO(R), female old age right hemisphere.

**Table 4 brainsci-11-00107-t004:** Subregion fractal dimension (FD) values in men of different ages.

Lobes	Index	MY (L)	MY(R)	MM(L)	MM(R)	MO(L)	MO (R)
Frontal Lobe	CMF	2.1162	2.1330	2.0865 ^**^	2.1400	2.0611 ^***^	2.1182
	FPol	2.3051	2.3125	2.2979	2.2922	2.3028	2.2943
	LOrF	2.3086	2.2961	2.2991	2.2766 ^**^	2.3063	2.2801 ***
	MOrF	2.2320 *	2.2634	2.2404 ^**^	2.2578	2.2419	2.2481
	Op	2.1858	2.2054	2.1668 ^**^	2.1889	2.1616	2.1760
	Or	2.2600	2.2214 *	2.2591	2.2181 ^**^	2.2588	2.2223 ^***^
	Tr	2.1825	2.1892	2.1669 ^**^	2.1936	2.1716 ^***^	2.2020
	RoMF	2.2646 *	2.2981	2.2424 ^**^	2.2744	2.2351 ^***^	2.2687
	SF	2.3363	2.3204	2.3208	2.3189	2.3158	2.3100
	PreC	2.1340	2.1090	2.1346	2.1203	2.1413	2.1136 ^***^
Limbic Lobe	CACg	2.4284	2.4255	2.4098	2.4115	2.4027	2.4061
	RoACg	2.4190	2.4262	2.4154	2.4047 ^**^	2.4108	2.4042
	IstCg	2.0524	2.0939	2.0351 ^**^	2.0714	2.0343 ^***^	2.0777
	Ins	2.2578	2.2412	2.2417	2.2303 ^**^	2.2357	2.2237
	PaH	2.1893	2.1970	2.1758	2.1635	2.1754	2.1656
	PoCg	2.1118	2.0722	2.1116	2.0920 ^**^	2.0892	2.0936
Temporal Lobe	B	2.3176	2.3096	2.2938	2.2920	2.2967	2.2891
	En	2.3509	2.3434	2.3311	2.3269	2.3304	2.3179
	IT	2.3146	2.3317	2.3130 ^**^	2.3273	2.3076	2.3156
	MT	2.0693	2.0542	2.0662	2.0356 ^**^	2.0538	2.0306 ^***^
	ST	2.3387	2.3501	2.3278 ^**^	2.3418	2.3211	2.3322
	TPol	2.1996	2.1899	2.1985	2.1925	2.1904	2.1808
	TrT	2.0899	1.9919 *	2.0543	1.9845 ^**^	2.0414	1.9616 ^***^
Parietal Lobe	IP	2.4268 *	2.4455	2.4035 ^**^	2.4342	2.3962 ^***^	2.4298
	PaC	2.1167	2.1198	2.1207	2.1036 ^**^	2.1047	2.0924
	PoC	2.2906	2.2840	2.2842	2.2769	2.2816	2.2633 ^***^
	PreCu	2.2137	2.2038	2.1912 ^**^	2.2106	2.1789	2.1963
	SP	2.3510	2.3432	2.3283 ^**^	2.3380	2.3255	2.3225
	SM	2.3697	2.3744	2.3526 ^**^	2.3602	2.3421	2.3517
Occipital Lobe	PerCa	2.3925	2.3813	2.3923	2.3739 ^**^	2.3852	2.3714 ^***^
	Fu	2.1176	2.1530	2.1013 ^**^	2.1455	2.1112 ^***^	2.1469
	Cu	2.3717	2.3807	2.3666 ^**^	2.3752	2.3648	2.3693
	LO	2.2782	2.2669	2.2705	2.2611	2.2766	2.2761
	Lg	2.0575	2.0531	2.0652 ^**^	2.0961	2.0950	2.1152
Number of significant smaller sub-regions		3	2	15	9	6	7

* Significantly smaller FD value (*p* < 0.05, FDR corrected) subregion of male young age group; ** significantly smaller FD value (*p* < 0.05, FDR corrected) subregion of male middle age group; *** significantly smaller FD value (*p* < 0.05, FDR corrected) subregion of male old age group. B, bankssts; CACg, caudal anterior cingulate; CMF, caudal middle frontal; Cu, cuneus; En, entorhinal; FPol, frontal pole; Fu, fusiform; IP, inferior parietal; IT, inferior temporal; Ins, insula; IstCg, isthmus cingulate; LO, lateral occipital; LOrF, lateral orbitofrontal; Lg, lingual; MOrF, medial orbitofrontal; MT, middle temporal; Or, pars orbitalis; Op, pars opercularis; Tr, pars triangularis; PaC, paracentral; PaH, parahippocampal; PerCa, pericalcarine; PreC, precentral gyrus; PreCu, precuneus; PoC, postcentral; PoCg, posterior cingulate; RoACg, rostral anterior cingulate; RoMF, rostral middle frontal; SF, superior frontal; SM, supra marginal; SP, superior parietal; ST, superior temporal; TPol, temporal pole; TrT, transverse temporal. MY(L), male young age left hemisphere; MY(R), male young age right hemisphere; MM(L), male middle age left hemisphere; MM(R), male middle age right hemisphere; MO(L), male old age left hemisphere; MO(R), male old age right hemisphere.

**Table 5 brainsci-11-00107-t005:** Significant decrease in fractal dimension (FD) values in cortical subregions from young to middle and middle to old age.

**From Young to Middle Age Period**	
**Left Hemisphere**	**Right: Hemisphere**
Frontal: Rostral middle frontal, Pars triangularis Temporal: Inferior temporal	Frontal: Superior frontal Parietal: paracentral
**From Middle to Old Age Period**	
**Left Hemisphere**	**Right Hemisphere**
Frontal: Rostral middle frontal, Limbic: Isthmus cingulate, Insula Posterior cingulate Temporal: Middle temporal, Superior temporal, Transverse temporal, Parietal: Inferior parietal, Supra marginal,Superior parietal Occipital: Lateral Occipital	Frontal: Rostral middle frontal, Precentral Limbic: Caudal anterior cingulate Temporal: Inferior temporal, Middle temporal Superior temporal, Transverse temporal, Parietal: Postcentral, Precuneus, Superior parietal Occipital: Lateral Occipital

**Table 6 brainsci-11-00107-t006:** Significant changes in cortical subregions at two different age periods for men and women.

**Female**	**Young to Middle Age Period**	
	**Left Hemisphere**	**Right Hemisphere**
	Frontal: Pars triangularis, Pars orbitalis Temporal: Infer temporal	Limbic: Posterior cingulate Temporal: Superior temporal Occipital: Lingual
	Middle to Old Age Period	
	Left:	Right
	Frontal: rostral middle frontal, caudal middle frontal Occipital: Lateral occipital	Frontal: caudal middle frontal Temporal: Transverse temporal Occipital: Lateral Occipital
**Male**	**Young to Middle Age Period**	
	**Left Hemisphere**	**Right Hemisphere**
	Frontal: Rostral middle frontal, Precentral Limbic: Insula Temporal: Inferior Temporal Parietal: Inferior parietal, Precuneus, Superior parietal, Transverse temporal Occipital: Fusiform	Frontal: Caudal middle frontal, Lateral orbitofrontal, Rostral middle frontal, Pars triangulars, Superior Frontal Occipital: Fusiform
	**Middle to Old Age Period**	
	**Left Hemisphere**	**Right Hemisphere**
	Frontal: Rostral middle frontal Temporal: Entorhinal cortex Parietal: Superior parietal	Frontal: precentral Temporal: middle temporal, Superior temporal, Transverse temporal Parietal: Postcentral, precuneus

## Data Availability

The data are not publicly available due to the privacy concern raised by our IRB.

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
