# Peer review of "Using Fractal Dimension Analysis with the Desikan–Killiany Atlas to Assess the Effects of Normal Aging on Subregional Cortex Alterations in Adulthood"

_brainsci, 2021, doi:10.3390/brainsci11010107_

Round 1

Reviewer 1 Report

The work of Chi-Wen and collegues aimed to described age and sex differences on functional and structural alterations in the human brain. The results appear interesting, but the structure of the experimental design does not allow to consider it as a work on sex-gender differences. There is no clear indication of the populations enrolled, in particular in women where, not knowing the phases of the menstrual cycle, of any oral contraceptive or HRT, of the state of menopause, it is not possible to establish whether these sex differences are only due to age or also related to hormonal status. therefore the study should be reviewed from a gender perspective, analyzing and dividing women based on their specific characteristics. Furthermore, novelty should be better highlighted

Introduction: Line 51-53. the specific effects of sexual hormones, menstrual cycle and
menopause should be reported. here some interesting references on the topic:
Arélin et al. Front Neurosci. 2015; 9: 44; Sex Med 2019;7:480e488; NeuroImage 220 (2020) 117125
Front Neurosci. 2014 Nov 24;8:380; Brain Sci. 2020 Apr; 10(4): 198; Clin Obstet Gynecol. 2008 September ; 51(3): 618–626

a recent paper published in Brain Imaging and Behavior (2020) 14:1831–1839: Gender differences in cortical morphological networks.
appears similar in content. What are the differences with the work presented here? Line 112:how were the ranges chosen? between 45 and 60 there can be
a mix of fertile, premenopausal and postmenopausal women.
do the authors know in which phase of the cycle the analyzes were carried out?
Are there women who took oral contraceptives, or HRT (and therefore in menopause)?
which differences in middle and old women? were all analysed during menopause?
it is not clear how males versus females were compared.

Author Response

Authors’ response to the comments of reviewer 1

Comment 1

The work of Chi-Wen and colleagues aimed to described age and sex differences on functional and structural alterations in the human brain. The results appear interesting, but the structure of the experimental design does not allow to consider it as a work on sex-gender differences. There is no clear indication of the populations enrolled, in particular in women where, not knowing the phases of the menstrual cycle, of any oral contraceptive or HRT, of the state of menopause, it is not possible to establish whether these sex differences are only due to age or also related to hormonal status. therefore, the study should be reviewed from a gender perspective, analyzing and dividing women based on their specific characteristics. Ffurthermore, novelty should be better highlighted

Author’s response: We thank the reviewer’s comments and valuable suggestions. In our work we aimed to use FD measure with DK atlas to assess normal aging effect on subcortical regions along different stages of adulthood. Although we divided the study groups into male and female groups, and we anticipated that men and women may reveal different morphological alterations as indicated by FD values along stages of adulthood, sex difference was not the main interest of our manuscript. Therefore, we thought that the title of our manuscript may cause such misunderstanding. Here, we revised the title of our manuscript as” Using Fractal Dimension Analysis with the Desikan–Killiany Atlas to Assess the Effects of Normal Aging on Subregional Cortex Alterations in Adulthood”. We hope the revised title can be more appropriate and suitable for presenting the aim of our revised manuscript. We have also cited the suggested references and added paragraphs in the Introduction and Discussion sections to explain how hormones affect brain structure and function along adulthood (please refer to the response below).  

Comment 2

Introduction: Line 51-53. the specific effects of sexual hormones, menstrual cycle and
menopause should be reported. here some interesting references on the topic:
1) Arélin et al. Front Neurosci. 2015; 9: 44; 2) Sex Med 2019;7:480e488; NeuroImage 220 (2020) 117125 3)Front Neurosci. 2014 Nov 24;8:380; 4) Brain Sci. 2020 Apr; 10(4): 198; 5) Clin Obstet Gynecol. 2008 September ; 51(3): 618–626

Authors’ response: we thank the reviewer’s valuable suggestions. We have surveyed the suggested papers and cited them into our references. In the revised manuscript, we have added a paragraph in the Introduction section to explain the effects of menstrual cycle and menopause on brain morphological and functional alteration in women. We have also added a paragraph in the Discussion section to clarify the limitation of our study that the effect of hormones on cortical morphological alterations cannot be assessed, especially for the female subjects. We have also deleted the term “sex difference” in the key words list. The added paragraph is labeled in bold-face type as follows:

In introduction section:

Numerous studies revealed that male and female showed different stages and cortical regions of brain atrophy with aging [18, 19]. …Elderly men exhibited more atrophy in the left hemisphere, whereas the atrophy was symmetric in elderly women [19]. “Besides normal aging, sex hormones also have a significant influence on alterations of brain structure, behavior, and cognitive functioning [20-23]. Postmenopausal women showed significantly decreased gray matter volumes of the insula, putamen, parahippocampal gyrus, amygdala, and anterior cingulate gyrus. These decreased regions were associated with decreased functional activity during visual sexual arousal in postmenopausal women [24]. For female across the menstrual cycle, intrinsic fluctuations in progesterone were associated with volumetric changes in CA2/3 of middle temporal lobe, entorhinal, perirhinal, and parahippocampal cortex [25].”

In discussion section:

“However, there are some limitations in the present study. The phases of menstrual cycle, taking oral contraceptives and state of menopause period of each female subject had not been inquired and investigated. These data are causal for assessing sex difference of normal aging. Hence in this study, the effect of hormone states on subcortical alteration along normal aging cannot be assessed, especially for the middle age female group who are in stages of menopausal transition.”

Added References:

  1. Arélin Katrin, Mueller Karsten, Barth Claudia, Rekkas Paraskevi V., Kratzsch Jürgen, Burmann Inga, Villringer Arno, Sacher Julia. Progesterone mediates brain functional connectivity changes during the menstrual cycle—a pilot resting state MRI study, Frontiers in Neuroscience,2015,9,44, DOI=10.3389/fnins.2015.00044
  2. Le J, Thomas N, Gurvich C. Cognition, The Menstrual Cycle, and Premenstrual Disorders: A Review. Brain Sci. 2020 Mar 27;10(4):198. doi: 10.3390/brainsci10040198. PMID: 32230889; PMCID: PMC7226433
  3. Nguyen, T. V., McCracken, J. T., Ducharme, S., Cropp, B. F., Botteron, K. N., Evans, A. C., & Karama, S. (2013). Interactive effects of dehydroepiandrosterone and testosterone on cortical thickness during early brain development. The Journal of neuroscience : the official journal of the Society for Neuroscience33(26), 10840–10848. https://doi.org/10.1523/JNEUROSCI.5747-12.2013
  4. Henderson VW. Cognitive changes after menopause: influence of estrogen. Clin Obstet Gynecol. 2008 Sep;51(3):618-26. doi: 10.1097/GRF.0b013e318180ba10. PMID: 18677155; PMCID: PMC2637911
  5. Baek HS, Kim GW, Sundaram T, Park K, Jeong GW. Brain Morphological Changes With Functional Deficit Associated With Sexual Arousal in Postmenopausal Women. Sex Med. 2019;7(4):480-488. doi:10.1016/j.esxm.2019.06.013
  6. Caitlin M. Taylor, Laura Pritschet, Rosanna K. Olsen, Evan Layher, Tyler Santander, Scott T. Grafton, Emily G. Jacobs, Progesterone shapes medial temporal lobe volume across the human menstrual cycle, NeuroImage, Volume 220, 2020, 117125, ISSN 1053-8119, https://doi.org/10.1016/j.neuroimage.2020.117125.

Comment 3

A recent paper published in Brain Imaging and Behavior (2020) 14:1831–1839: Gender differences in cortical morphological networks. appears similar in content. What are the differences with the work presented here?

Authors’ response: We appreciate the reviewer’s valuable suggestions. In the suggested paper of Ahmed Nebli and Islem Rekik (please refer to Figure 3-1 below), the authors used cortical morphological network with machine learning technology to assess the gender difference between male and female group (age from 21 to 23 years old, please refer to Figure 3-2). First they used DK atlas to paecellate the gray matter cortex into sub-regions and measure four morphological indexes of the parcellated sub-regions including maximum principal curvature, cortical thickness, sulcal depth, and average curvature to calculate their correlation matrix and using graphic theorem to built cortical network of each measure. They used these networks with machine learning technology to assess discriminative morphological connection between genders. The whole analysis procedures of “Brain Imaging and Behavior (2020) 14:1831–1839” was summarized in Figure 3-4.

Whereas in our study, we used 3D-FD analysis with the Desikan–Killiany (DK) atlas to assess subregional morphological changes in adulthood. A total of 258 participants (112 women and 146 men) aged 30–85 years old participated in this study. In this study, we hypothesize different degrees of morphological alterations in cortical subregions across different stages of adulthood to reveal the normal aging effect on morphological changes. We aimed to assess whether FD values are a precise and superior predictor for aging effect, and those morphological alterations in the subregions manifest as functional impairment across different stages of adulthood in normal aging. We concluded that our FD analysis using the DK atlas-based prediagnosis may provide a suitable tool for assessing normal aging and neurodegeneration between groups or in individual patients. Figure 3-5 illustrates the analysis procedure of our study. We have summarized the differences between our work and Brain Imaging and Behavior (2020) 14:1831–1839, and illustrated them in Table 3-1 below.

Table 3-1 Differences between “Brain Imaging and Behavior (2020) 14:1831–1839” and Manuscript ID: brainsci-1039742 of Chi-Wen Jao

Differences

Ahmed Nebli and Islem Rekik

Chi-Wen Jao and colleagues

Study groups

308 male and 390 female, age 21-23 years old

146 male and 112 female, aged 30–85 years old

Measure method

Maximum principal curvature, cortical thickness, sulcal depth, and average curvature to build cortical morphological networks (CMN)

Fractal dimension values of subregions of cortex( parcellated by using DK atlas)

Comparison

Cortical morphological networks (CMN) between male and female groups

FD values of parcellated subregions between different age period groups of male and female

Conclusions

 The most discriminative CMN connections between males and females were derived from the left hemisphere using the mean sulcal depth as measurement. The first most discriminative morphological connection revealed across all cortical attributes involved (entorhinal cortex ↔ caudal anterior cingulate cortex) and (entorhinal cortex ↔ transverse temporal cortex) respectively, which gives us new insights into behavioral gender differences from an omics perspective and might explain why males and females learn differently.

Men exhibited more decrease in FD values in the subregions on the left than those in the right, whereas women exhibited more decrease in the lateral subregions. Older men were at a higher risk of developing mild cognitive impairment (MCI) and exhibited age-related memory decline earlier than women. Our FD analysis using the DK atlas-based prediagnosis may provide a suitable tool for assessing normal aging and neurodegeneration between groups or in individual patients.

Figure 3-1 Abstract of Brain Imaging and Behavior (2020) 14:1831–1839

Figure 3-2                                  Figure 3-3

Figure 3-4                                 Figure 3-5

Comment 4

Line 112: How were the ranges chosen? between 45 and 60 there can be a mix of fertile, premenopausal and postmenopausal women.do the authors know in which phase of the cycle the analyzes were carried out?

Authors’ response: We thank the reviewer’s suggestions. According to the definition of older person in most developed world country from WHO (please refer to Figure 4-1 below), those who are older than 65 years old can be regarded as older persons. One paper from BMJ published in 1997 (please refer to figure 4-2), the authors defined the age range of a middle age person was from 40 to 60 years old. Another paper of Jungae Kim (International Journal of Advanced Culture Technology Vol.8 No.2 18-27 (2020), Figure 4-3), defined the age range of middle aged people was from 45 to 65 years old. Another paper defined the age range of middle age as from 40 to 60 years old (Neuropsychiatric disease and treatment, 2016, 12, 453–465, Figure 4-4). According to these age-related studies, in this study we thought the age period from 45 years to 60 years is appropriate for the age range of middle age group. In the revised manuscript, we have cited these papers into our references and the revised sentence in the Material and Method Section as follows (labeled in bold-face type):

2.1. Participants

A total of 258 healthy adults …and no neurological abnormalities during the study. “Referring to the age stages of previous studies, the young, middle, and old adulthood groups included individuals less than 45, from 45 to 60, and more than 60 years old, respectively [?].

Figure 4-1

Figure 4-2

Figure 4-3

Figure 4-4

Added references:

  1. Shaper, A. G., Wannamethee, S. G., & Walker, M. (1997). Body weight: implications for the prevention of coronary heart disease, stroke, and diabetes mellitus in a cohort study of middle aged men. BMJ (Clinical research ed.)314(7090), 1311–1317. https://doi.org/10.1136/bmj.314.7090.1311
  2. Kim, J. (2020). Effect of Subjective Health Perception and Mental Health Status on the Quality of Life in the Late Middle Age. International Journal of Advanced Culture Technology8(2), 18–27. https://doi.org/10.17703/IJACT.2020.8.2.18
  3. Liu, H., Wang, L., Geng, Z., Zhu, Q., Song, Z., Chang, R., & Lv, H. (2016). A voxel-based morphometric study of age- and sex-related changes in white matter volume in the normal aging brain. Neuropsychiatric disease and treatment12, 453–465. https://doi.org/10.2147/NDT.S90674

Comment 5

Are there women who took oral contraceptives, or HRT (and therefore in menopause)? which differences in middle and old women? were all analysed during menopause? it is not clear how males versus females were compared.

Author’s response: we appreciate the reviewer’s comments. In this study, we haven’t inquired the menstrual cycle, taking oral contraceptives and state of menopause period of each female. We thank the reviewer’s valuable suggestions that will greatly improve the quality of our work. Therefore, to avoid that the title of our original manuscript may cause misunderstanding. Here, we revised the title of our manuscript as” Using Fractal Dimension Analysis with the Desikan–Killiany Atlas to Assess the Effects of Normal Aging on Subregional Cortex Alterations in Adulthood”. We hope the revised title can be more appropriate and suitable for presenting the aim of our revised manuscript. We have also added a paragraph in the Discussion section to clarify the limitation of our study that the effect of hormones on cortical morphological alterations cannot be assessed, especially for the female subjects. We have also deleted the term “sex difference” in the key words list. The added paragraph is labeled in bold-face type as follows:

In discussion section:

“However, there are some limitations in the present study. The phases of menstrual cycle, taking oral contraceptives and state of menopause period of each female subject had not been inquired and investigated. These data are causal for assessing sex difference of normal aging. Hence in this study, the effect of hormone states on subcortical alteration along normal aging cannot be assessed, especially for the middle age female group who are in stages of menopausal transition.”

Reviewer 2 Report

The paper presents results on using the fractal dimension to quantify morphological changes in the brain related to aging with a particular focus on sex differences.

The results are sound and clearly presented. However, there is some space for improvement.

1. I think the presentation can be improved - see below for the details. In particular I found parts of the introduction redundant. Moreover, I would find it useful, if the authors would clearly indicate, when by whom the method of using the FD to quantify morphological changes in the brain was introduced and what is the new contribution of the presented paper. I am not an expert in the field, but from a short literature search I found several papers that use the method of FD in the same context of age related changes and that are not cited, for instance papers by Christopher R. Madan and coworkers (Cortical complexity as a measure of age-related brain atrophy, Neuroimage 134 (2016), 617-629, Age-related decrements in cortical gyrification: Evidence from an accelerated longitudinal dataset Eur J Neurosci 2020) or C. Marzi et al., Toward a more reliable characterization of fractal properties of the cerebral cortex of healthy subjects during the lifespan, Scientific reports (2020) 10:16957.

2. The fractal dimension is presented as a measure of complexity. This is a widespread view, which is, however, problematic at last in two aspects:
a) Complexity in general is not well defined and therefore a vague concept. Thus, it is not clear, what the complexity of brain morphology should be beyond its fractal dimension. But then, one also could avoid using the word "complexity". Because
b) according to its fractal dimension a ball (D=3) is more complex than a sphere (D=2), but, a strange attractor like the Lorenz attractor (D approximately 2.1) is less complex than a ball. However, many people would probably call the structure of the Lorenz attractor more complex than the structure of a ball.

3. The results are presented in tables reporting the group averages. How are the results distributed? Can one expect the same lateralization for the whole group? What about the handedness of the subjects?

More detailed remarks:

p.1 l.35: The authors write "Fractal dimension (FD) is a quantitative measure of cerebral folding." I find this sentence a bit strange, because first fractal dimension is a mathematical concepts to characterize fractal sets or measures, respectively. Perhaps one can write: "The concept of fractal dimension turned out to be a useful measure to quantify cerebral folding." or something similar, depending on how the authors relate their study to previous studies employing the concept of fractal dimension changes in brain structure.

p.2 l.75: There is some redundancy here. The authors write "Men were reported to exhibit more brain atrophy with aging that did woman [18]" while the previous sentences reported already more specific results from the same source [18].

p.2 l. 78-80: Seems to be at least partially redundant with the previous paragraphs.

p.2 l.81: "larger variances and more sex effect" "larger" and "more" compared to what?

p.2 l.88: "Because FD analysis is based on a logarithmic scale, even a small increase 88 in FD value corresponds to a considerable increase in complexity; hence, a higher FD value represents 89 greater topological complexity of the tissue under study". I find the statement tautologic.

p2. l.95-97; There is some overlap with the Materials and Methods section here.

p4. l.145/146: There are different ways to define fractal dimensions (see for instance the book by Kenneth Falconer, Fractal Geometry). The Hausdorff dimension is one way, the box-counting dimension another. However, box-counting dimension and capacity are only different names for the same dimension.

p.4 l. 158/159: Same sentence as on p.2 l.88. See also my remark above.

p.5 l. 171: Is the four-digit precision for the dimension estimates justified? I would doubt that.

Author Response

Authors’ response to comments of reviewer 2

Comment 1

The results are sound and clearly presented. However, there is some space for improvement.

  1. I think the presentation can be improved - see below for the details. In particular I found parts of the introduction redundant. Moreover, I would find it useful, if the authors would clearly indicate, when by whom the method of using the FD to quantify morphological changes in the brain was introduced and what is the new contribution of the presented paper. I am not an expert in the field, but from a short literature search I found several papers that use the method of FD in the same context of age related changes and that are not cited, for instance papers by Christopher R. Madan and coworkers (Cortical complexity as a measure of age-related brain atrophy, Neuroimage 134 (2016), 617-629, Age-related decrements in cortical gyrification: Evidence from an accelerated longitudinal dataset Eur J Neurosci 2020) or C. Marzi et al., Toward a more reliable characterization of fractal properties of the cerebral cortex of healthy subjects during the lifespan, Scientific reports (2020) 10:16957.

Author’s response: we appreciate the reviewer’s comments and suggestions; their suggestions are valuable and can intensively improve our work more comprehensive and easy to read. In our revised manuscript, we have deleted the redundant and overlapped sentences to make the introduction briefer and more comprehensive. We thank the reviewer for suggesting important references of FD studies of aging related cortical changes. In the revised manuscript, we have added the suggested papers in our reference list and have explain the highlights of these papers in Materials and Methods section. The revised paragraph is as follows:    

In Materials and Methods section

2.3. Three-Dimensional FD Measurement of Parcellated Regions

A fractal is an irregular object in which part of the compartment resembles the overall shape. Kiselev et al. demonstrated the cerebral cortex had the self-similar fractal structure and its FD value was 2.80± 0.05 [38]. Shan et al. suggested that the FD index is a compact measure of structural complexity, and can be used as a summary index of structure irregularity [39].”FD has been extensively used to investigate morphological changes caused by neurological diseases in the cerebral cortex [16–18]. “For assessing cortical alteration in aging, FD has been reported that was more strongly related to age than other cortical measures [40]. Recently, Marzi et al. demonstrated that FD revealed strong associations with age and yielded the most accurate machine learning models for individual age prediction [41]”.

Added references:

  1. Kiselev, VG, Hahn, KR, Auer, DP, 2003. Is the brain cortex a fractal? NeuroImage 20, 1765–1774.
  2. Shan, Z.Y., Liu, J.Z., Glassa, J.O., Gajjarc, A., Lid, C.S., Reddicka, W.E., 2006. Quantitative morphologic evaluation of white matter in survivors of childhood medulloblastoma. Magn. Reson. Imaging 24, 1015–1022.

  1. Christopher R. Madan, Elizabeth A. Kensinger, Cortical complexity as a measure of age-related brain atrophy, NeuroImage, Volume 134, 2016, Pages 617-629, ISSN 1053-8119, https://doi.org/10.1016/j.neuroimage.2016.04.029

  1. Marzi, C., Giannelli, M., Tessa, C. et al.Toward a more reliable characterization of fractal properties of the cerebral cortex of healthy subjects during the lifespan. Sci Rep10, 16957 (2020). https://doi.org/10.1038/s41598-020-73961-w

.

Comment 2

  1. The fractal dimension is presented as a measure of complexity. This is a widespread view, which is, however, problematic at last in two aspects:
    a) Complexity in general is not well defined and therefore a vague concept. Thus, it is not clear, what the complexity of brain morphology should be beyond its fractal dimension. But then, one also could avoid using the word "complexity". Because
    b) according to its fractal dimension a ball (D=3) is more complex than a sphere (D=2), but, a strange attractor like the Lorenz attractor (D approximately 2.1) is less complex than a ball. However, many people would probably call the structure of the Lorenz attractor more complex than the structure of a ball.

Authors’ response: We appreciate the reviewer’s suggestion. Basically, a fractal object is that its geometric figure is self-similar if there is a point where every neighborhood of the point contains a copy of the entire figure. Like figure 2-1 illustrates, it’s a combination of different size of square diagram, it’s a self-similar figure from partial to all and it’s a pure fractal object. Precisely, a ball is not a pure fractal nor the sphere but the Lorenz attractor is really a fractal object. Similarly, the brain cortex is not a pure fractal, and for some scales range the brain cortex can be measured as a fractal. Kiselev at al. confirmed the fractal nature of the human cerebral cortex down to a spatial scale of 3 mm (please refer to Figure 2-2). They also confirmed the brain cortex can be regarded as a fractal from spatial scale of 2mm to 20mm (please refer to Figure 2-3). In our work we also have confirmed from spatial scale of 2mm to 8mm, the gray cortex can be regard as a fractal. We appreciate the reviewer’s suggestion that avoided using the term “complexity” which may cause vague concept to the readers. We are all highly agreed with their suggestion. In one of our previous publication we named FD value of cortex folding as “cortical complexity” (please refer to Figure 2-4). We think the term “cortical complexity” is more appropriate than “morphological complexity” to indicate the complexity of cortical folding. Hence, in the revised manuscript, we replaced the term “morphological complexity” with “cortical complexity”.     

Figure 2-1

Figure 2-2

Figure 2-3

Figure 2-4

Comment 3

  1. The results are presented in tables reporting the group averages. How are the results distributed? Can one expect the same lateralization for the whole group? What about the handedness of the subjects?

Authors’ response: We thank the reviewer’s comment. In one of our previous FD study publication, we reported that FD measure has a smaller variance. (please refer to Figure 3-1). For exampling the distribution of FD vales in each subregion, we have plot the distribution of FD values of left CMF (caudal middle frontal) of male middle age and female middle age groups (male: please refer to Figure 3-2, female: please refer to Figure 3-3). We also calculated the standard variance of these two FD values, and for left CMF of middle age male is 0.0895, and 0.0921 for female middle age group. The ratio of standard variance to the mean FD value is 4.29% for male, and 4.45% for female. From Figure3-2 and 3-3, the distribution diagram of left CMF for female and male group reveal a normal Gaussian distribution, and the FD values of female and male in left CMF have smaller variance. The lateralization between two hemispheres indicated the tendency of subregional alterations of group during aging, but for an individual subject, we think there is some individual difference in lateralization phenomenon, and hence can’t expect one’s lateralization from the lateralization result of the whole group. In this study, the handedness of each subject was not inquired. We know that hand preference may influenced regional asymmetries in parietal association and dorsomedial frontal cortices. The percent rate of left hand preference in our country is about 5%, and the subject are not easy to recruit. We may include this issue in our future study.

Figure 3-1

Figure 3-2                             Figure 3-3

Comment 4

More detailed remarks:

p.1 l.35: The authors write "Fractal dimension (FD) is a quantitative measure of cerebral folding." I find this sentence a bit strange, because first fractal dimension is a mathematical concept to characterize fractal sets or measures, respectively. Perhaps one can write: "The concept of fractal dimension turned out to be a useful measure to quantify cerebral folding." or something similar, depending on how the authors relate their study to previous studies employing the concept of fractal dimension changes in brain structure.

Authors’ response: we thank the reviewer’s suggestion. In the revised manuscript, we have the sentence and the revised manuscript is as follows (in boldface type): 

In abstract section:

Abstract: Normal aging is associated with functional and structural alterations in the human brain. The effects of normal aging and genders on morphological changes in specific regions of the brain are unknown. “Fractal dimension (FD) can be a quantitative measure of cerebral folding.”

Comment 5

p.2 l.75: There is some redundancy here. The authors write "Men were reported to exhibit more brain atrophy with aging that did woman [18]" while the previous sentences reported already more specific results from the same source [18].

Authors’ response: We thank reviewer’s comment. In the revised manuscript, we have deleted the sentence “Men were reported to exhibit more brain atrophy with aging that did woman [18]"

Comment 6

p.2 l. 78-80: Seems to be at least partially redundant with the previous paragraphs.

Authors’ response: We appreciate reviewer’s comment. In the revised manuscript, we shorten the paragraph and deleted redundant sentences, the revised paragraph in the Introduction section is as follows:

In Introduction section:

“Studies have suggested that men and women showed different brain atrophy with aging [18, 19]. Moreover, the degree of brain atrophy was milder in women than in men with aging; however, the opposite results have also been reported [18]. After controlling for total percent brain volume change, men showed greater relative regional brain reduction than women in bilateral precentral gyri, bilateral paracingulate gyri, and bilateral supplementary motor cortices [20]. Sex differences may affect lateralization of brain atrophy with aging. Elderly men exhibited more atrophy in the left hemisphere, whereas the atrophy was symmetric in elderly women [19].”

Comment 7

p.2 l.81: "larger variances and more sex effect" "larger" and "more" compared to what?

Authors’ response: We thank reviewer’s suggestion. In the revised manuscript, we have revised the sentence as follows:

As compared with cortical thickness, these conventional volumetric methods may reveal larger variances and more sex effect [24].”

Comment 8

p.2 l.88: "Because FD analysis is based on a logarithmic scale, even a small increase in FD value corresponds to a considerable increase in complexity; hence, a higher FD value represents greater topological complexity of the tissue under study". I find the statement tautologic.

Authors’ response: We thank reviewer’s comment and suggestion. We have delete this sentence in the revised manuscript

Comment 9

p2. l.95-97; There is some overlap with the Materials and Methods section here.

Authors’ response:

We have delete some sentence in the introduction to make it more brief and not overlap with materials and methods section, please refer to our revised manuscript.

Comment 10

p4. l.145/146: There are different ways to define fractal dimensions (see for instance the book by Kenneth Falconer, Fractal Geometry). The Hausdorff dimension is one way, the box-counting dimension another. However, box-counting dimension and capacity are only different names for the same dimension.

Authors’ response: We thank reviewer’s comment. We have revised the sentence as follows:

“FD is a quantitative indicator of object complexity (or irregularity), which can be calculated on the basis of different metrics, namely the Hausdorff dimension and box-counting dimension.”

Comment 11

p.4 l. 158/159: Same sentence as on p.2 l.88. See also my remark above.

Authors’ response: We thank reviewer's suggestion, and We have deleted the same sentence on p.2 1.88

Comment 12

p.5 l. 171: Is the four-digit precision for the dimension estimates justified? I would doubt that.

Authors’ response: We thank reviewer’s comment. The FD value of each subregion was calculated from the slope of the line from poly fit analysis. We have replicated the FD measure procedure and illustrated in Figure 12-1 below. As indicated in Figure 12-1, the FD value is 2.2962 and 2.2666  (please refer to the red rectangular in Figure 12-1)from poly fit results. 

Reviewer 3 Report

The authors presented new method for assessing effects of aging and sex on alterations in subcortical regions. The topic is very interesting because it is directed  towards a personalized approach to the patient, which is a tendency in  modern medicine.

The manuscript is designed in accordance to the instructions for authors. Through the introduction authors indicate the limitations of existing methods and introduced their approach. The method and results are well presented. References correspond to the content of the text and there is no unnecessary self-citation. 

Because of the new approach to the method and complicated correlations between age and sex groups, the work presents a high publication value.

Only a few revisions are needed:

1. Page 4, line 133 – Change Table 1 to Table 2

2. Page 8, Figure 3 - Increase the font of sub-region names marked in Figures (a) and (b). The same remark applies to Figures 4 and 5 (Page 10).

3. Page 8, line 246-249 - Figure 3 title should be: Figure 3. Significantly decreased fractal dimension (FD) values ​​in cortical subregions between different age periods. OR  Significant decrease in fractal dimension (FD) values ​​in cortical subregions between different age periods. Further, it is enough to use abbreviation FD in the caption. The same remark applies to Figures 4 and 5 (Page 10, line 275-286).

4. Page 11-12, line 347-370 – Change 5a to 6a and 5b to 6b. In this section Figure 6a and 6b are described but in the text Figure 5a and 5b are used instead.

5. Page 12, line 374 – The title of Figure 6 is missing (for example: Prediagnosis system based on the fractal dimension (FD) measuring)  

Author Response

The authors presented new method for assessing effects of aging and sex on alterations in subcortical regions. The topic is very interesting because it is directed  towards a personalized approach to the patient, which is a tendency in  modern medicine. The manuscript is designed in accordance to the instructions for authors. Through the introduction authors indicate the limitations of existing methods and introduced their approach. The method and results are well presented. References correspond to the content of the text and there is no unnecessary self-citation.  Because of the new approach to the method and complicated correlations between age and sex groups, the work presents a high publication value.

Authors’ response:

We appreciate the reviewer’s comment and praise.

Only a few revisions are needed:

Comment 1

  1. Page 4, line 133– Change Table 1 to Table 2

Author’s response: We thank the reviewer’s suggestion. We have revised it.

Comment 2

  1. Page 8, Figure 3- Increase the font of sub-region names marked in Figures (a) and (b). The same remark applies to Figures 4 and 5 (Page 10).

Author’s response: We thank the reviewer’s reminder, and have revised it.

Comment 3

  1. Page 8, line 246-249- Figure 3 title should be: Figure 3. Significantly decreased fractal dimension (FD) values ​​in cortical subregions between different age periods. OR  Significant decrease in fractal dimension (FD) values ​​in cortical subregions between different age periods. Further, it is enough to use abbreviation FD in the caption. The same remark applies to Figures 4 and 5 (Page 10, line 275-286).

Author’s response: We thank the reviewer’s suggestion and have revised them.

Comment 4

  1. Page 11-12, line 347-370– Change 5a to 6a and 5b to 6b. In this section Figure 6a and 6b are described but in the text Figure 5a and 5b are used instead.

Author’s response: We thank the reviewer’s suggestion and have revised it.

Comment 5

  1. Page 12, line 374– The title of Figure 6 is missing (for example: Prediagnosis system based on the fractal dimension (FD) measuring)  

Author’s response: We thank reviewer’s suggestion, and have revised the title of Figure 6 as follows:

“Figure 6. Prediagnosis system based on FD with Desikan–Killiany atlas measuring (a) Participant 37 from…abnormal subregions
